# PISA: A Pragmatic Psych-Inspired Unified Memory System for Enhanced AI Agency

## Abstract

Memory systems are fundamental to AI agents, yet existing work often lacks adaptability to diverse tasks and overlooks the constructive and task-oriented role of AI agent memory. Drawing from Piaget's theory of cognitive development, we propose PISA, a pragmatic, psych-inspired unified memory system that addresses these limitations by treating memory as a constructive and adaptive process. To enable continuous learning and adaptability, PISA introduces a trimodal adaptation mechanism (*i.e.*, schema updation, schema evolution, and schema creation) that preserves coherent organization while supporting flexible memory updates. Building on these schema-grounded structures, we further design a hybrid memory access architecture that seamlessly integrates symbolic reasoning with neural retrieval, significantly improving retrieval accuracy and efficiency. Our empirical evaluation, conducted on the existing LOCOMO benchmark and our newly proposed AggQA benchmark for data analysis tasks, confirms that PISA sets a new state-of-the-art by significantly enhancing adaptability and long-term knowledge retention. The source code of PISA and data of AggQA are available at https://anonymous.4open.science/r/PISA-421/

## 1 Introduction

> *"Knowledge is not a copy of reality. To know an object is to act upon it and to transform it."*
> — *Jean Piaget*

In recent years, Large Language Model (LLM) based AI agents have been proposed to autonomously accomplish specific objectives or solve particular types of tasks (Xi et al., 2025; Liang et al., 2022; Hu et al., 2024; Zhang et al., 2025; Nathani et al., 2025; Wang et al., 2025; Fish et al., 2025). Their core characteristic lies in task-oriented behavior, which involves perceiving the environment, executing reasoning and planning, and implementing a series of coherent actions to achieve task objectives. During task execution, the rational utilization of historical information has been proven to be a crucial pathway for enhancing efficiency and effectiveness. However, indiscriminately inputting all historical information into the AI agent at each decision point introduces two implicit challenges: (1) excessive historical information both increases reasoning costs and risks exceeding the model's context window (Keles et al., 2023); (2) the substantial amount of irrelevant information contained therein may interfere with the model's judgment and decision-making (Wang & Sun, 2025).

Agent memory (Packer et al., 2024; Zhong et al., 2024; Wang et al., 2024; Xu et al., 2025; Kang et al., 2025; Chhikara et al., 2025; Rasmussen et al., 2025) has been proposed as a set of practical approaches by constructing retrieval knowledge bases through persistent storage and structured organization of historical experiences, thereby improving future decision quality and efficiency. Nevertheless, existing work predominantly focuses on engineering optimizations, *e.g.*, memory persistence and retrieval efficiency, neglecting the active constructive nature and the task-oriented characteristic of AI agents. Meanwhile, current agent memory benchmarks (Maharana et al., 2024a; Wu et al., 2025; Hu et al., 2025) are limited to effectiveness evaluation, with most focusing on contextualized dialogue for fact retrieval, lacking task-specific assessments (Rasmussen et al., 2025).

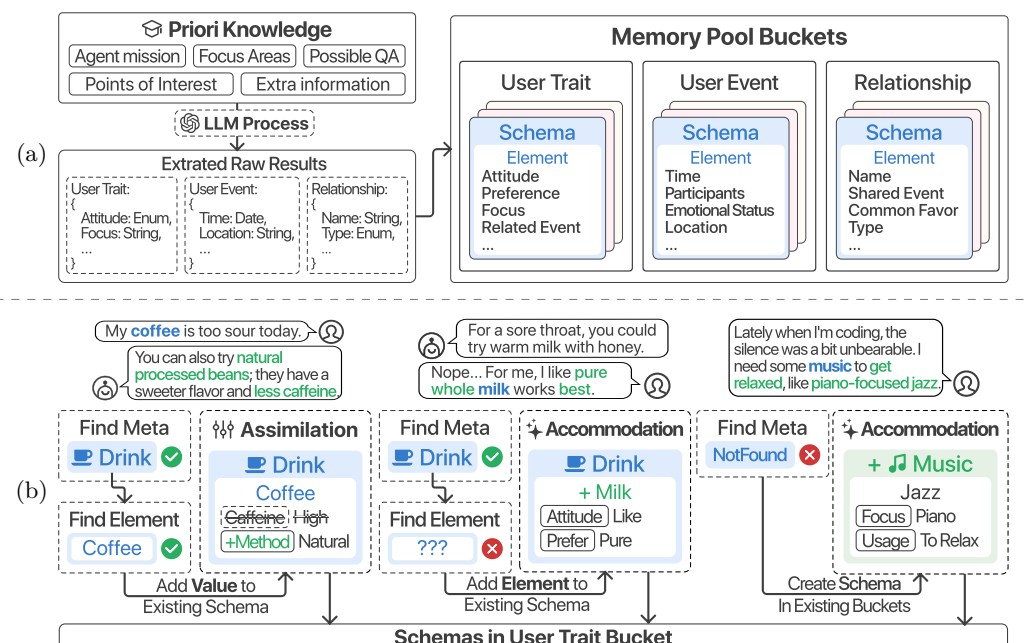

Figure 1: Example of PISA's two primary functions. (a) System initialization by analyzing prior knowledge to construct task-oriented memory buckets (*e.g.*, User Trait) with predefined schemas. (b) Dynamic memory evolution through a tri-modal mechanism: (1) Assimilation integrates new values into an existing schema (*e.g.*, coffee details); (2) Accommodation either evolves a schema by adding a new element (*e.g.*, "Milk"); or (3) creates an entirely new schema for novel concepts (*e.g.*, "Music").

Consequently, current memory systems exhibit poor performance on heterogeneous tasks, as demonstrated in our benchmarking results in Table 2. Inspired by Piaget's cognitive theory (Piaget et al., 1952) that knowledge is actively constructed through interaction with the environment rather than being passively absorbed, we strive to introduce a novel memory system that functions as an active and constructive process, enabling dynamic adaptation to task demands.

Building on this perspective, we propose *a pragmatic Psych-Inspired unified memory System for enhanced AI Agency*, named PISA, an actively constructive yet adaptive agent memory system that integrates differentiated architectures with flexible update strategies. PISA first constructs task-oriented memory structures by analyzing the objectives of the AI agent as illustrated in Figure 1a, thereby addressing the limitations of conventional structured memory that lacks adaptability to heterogeneous tasks. Informed by Piaget's schema theory[1], *i.e.*, cognitive development progresses through two complementary processes: assimilation, in which new experiences are integrated into existing schemas, and accommodation, in which schemas are modified or newly created to capture novel or conflicting information, we further introduce memory adaptation strategies for more flexibility. More specifically, we propose dynamic evolutionary mechanisms to model structural knowledge paradigms and semantic relationships. As shown in Figure 1b, our adaptive workflow module dynamically manages incoming experiences by assimilating them into existing schemas, accommodating schema evolution, or creating new schemas. This tri-modal approach ensures coherent knowledge organization, continuous learning, and adaptable memory updates. Building on this, we design a hybrid memory access architecture integrating symbolic reasoning with neural retrieval. Through extensive experiments on various models and benchmarks, our approach demonstrates significant improvements, consistently outperforming all baseline models across all evaluation metrics.

---

[1]Details are described in Appendix C.1

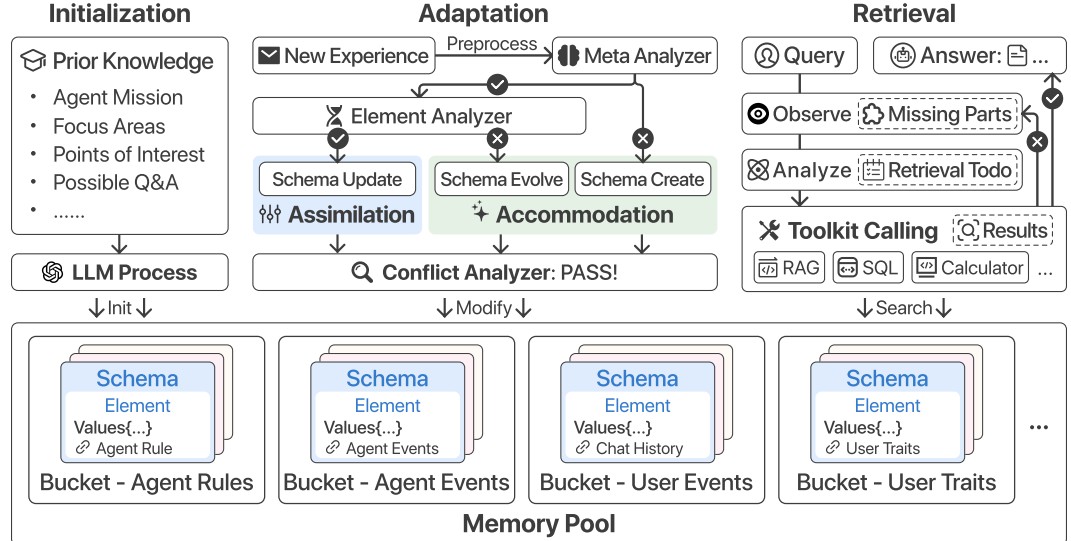

Figure 2: Overall architecture of PISA. The system is built upon a uniquely designed `Schema Engine` and consists of three core modules: (1) the Initialization Module, which leverages agent goals and prior knowledge to initialize memory structures; (2) the Adaptation Module, which dynamically processes new experiences through schema assimilation, evolution, and creation, guided by conflict analysis; and (3) the Retrieval Module, which interprets queries and retrieves relevant memories using task-aware retrieval strategies and external tool calls. Together, these components enable a flexible, extensible, and task-oriented memory system for LLM-based AI agents.

Overall, the contributions of this paper include the following aspects:

1. Unified Memory System Design: We propose a unified memory framework PISA based on Piaget's cognitive theory, that constructs task-oriented memory structures and overcomes the adaptability limitations of general-purpose memory systems.

2. Adaptive Memory Evolution Mechanism: We introduce a Piaget-inspired adaptive memory evolution mechanism, which enhances memory flexibility and enables continuous learning through a trimodal update process.

3. Hybrid Memory Access Architecture: We design a hybrid access mechanism based on task-oriented memory structures that integrates symbolic reasoning with neural retrieval, significantly improving the efficiency and accuracy of memory retrieval.

4. Real-world Scenario Evaluation Benchmark: We construct AggQA, a mini-benchmark for data analysis tasks spanning practical domains including Medical and Finance, to address the deficiencies in existing benchmarks regarding task diversity and practical relevance.

## 2 PISA: A Pragmatic Psych-Inspired Unified Memory System for Enhanced AI Agency

The overall architecture of PISA is illustrated in Figure 2. The entire system is built upon a uniquely designed Schema Engine and comprises three modules: the Initialization Module, the Adaptation Module, and the Retrieval Module. The Schema Engine serves as the core component of PISA, responsible for defining and managing the memory structure of the AI agent. The Initialization Module initializes PISA's memory structure based on the AI agent's goals and tasks. The Adaptation Module processes experiences, while the Retrieval Module retrieves relevant memories from PISA according to the AI agent's goals and tasks.

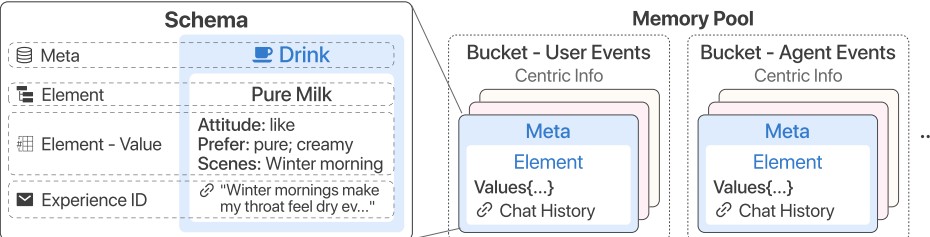

Figure 3: Schema Engine of PISA. The memory system is organized hierarchically into Memory Pool, Buckets, and the Schema. Each schema encodes structured knowledge through Meta, Element, Element-Value, and a unique experience ID, enabling flexible, extensible, and task-oriented memory organization.

In the following sections, we will provide a detailed introduction to the functionality and design of the Schema Engine and each module.

## 2.1 SCHEMA ENGINE

This part details our proposed memory architecture, as illustrated in Figure 3, which comprises a three-tiered hierarchy: the Memory Pool, Buckets, and the Schema. The Memory Pool acts as the comprehensive repository for all data, which is partitioned into thematic Buckets such as "Bucket - Agent Events" and "Bucket - User Events". Each bucket's Centric Info defines the core focus of the memories stored within and the principles for their structuring. The foundational unit of storage is the Schema. Each Schema is composed of four main components. The Meta defines a high-level topic (*e.g.*, "Drink"), under which a specific Element is identified (*e.g.*, "Pure Milk"). The Element-Value component then captures nuanced descriptors in a flexible key-value format implemented via "JSONB", such as "Attitude: like" or "Scenes: Winter morning". Crucially, a unique Experience ID links this structured Schema entry back to its originating unstructured event in the Memory Pool, such as a specific "Chat History" log. This design yields a memory system that is logically flexible and extensible, while maintaining a queryable, structured representation in its physical implementation.

## 2.2 INITIALIZATION MODULE OF PISA FOR AI AGENT GOAL

The Initialization Module leverages prior knowledge, particularly the agent's goals and task specifications, to proactively construct the memory structure of PISA. Rather than passively storing information, this module embeds task priors into the organization of the memory pool, ensuring that memory is oriented toward the agent's objectives from the outset. Concretely, given an agent goal, the module first analyzes its semantic requirements and determines the *centric info of buckets* (*e.g.*, user traits, user events, relationships), which provide coarse-grained partitions for categorizing incoming experiences. Within each bucket, the module further specifies the *schema-level element–value structures*, defining which attributes should be actively monitored (*e.g.*, time, location, emotional status), while leaving their values unpopulated until actual experiences are observed.

Through this hierarchical initialization process, PISA transforms memory into a task-sensitive scaffold that guides subsequent assimilation and retrieval, thereby reducing irrelevant information and enhancing the efficiency of downstream reasoning.

## 2.3 ADAPTATION MODULE

Building on Piaget's schema theory and the implemented Schema Engine, the Adaptation Module employs a dynamic evolutionary paradigm to capture both structural knowledge patterns and semantic relationships. This module enables PISA to autonomously determine whether new experiences should be assimilated into existing schemas, accommodated through schema evolution, or used to create entirely new schemas. This tri-modal mecha-

nism ensures continuous learning, maintains coherent organization, and addresses the flexibility challenges of memory updates. The detailed workflow is summarized in Algorithm 1, with the following sections elaborating on each component.

**Notation and Symbols.** We summarize the symbols used throughout this section:

| Symbol | Description |
|---|---|
| $b$ | Bucket index determined during preprocessing; $K_b$ is the canonical key set for bucket $b$. |
| $S(b)$ | Current set of schemas in bucket $b$; $\sigma \in S(b)$ denotes a schema with metadata $m(\sigma)$ and element set $U(\sigma)$. |
| $u$ | An element in $U(\sigma)$; $V(u)$ is the multiset of records attached to $u$. |
| $e$ | An incoming experience segment; $r(e)$ is the normalized record extracted from $e$ projected onto $K_b$; $m(e)$ is metadata extracted from $e$. |
| $s_{\mathrm{meta}}(e, \sigma)$ | Schema-level similarity score in $[0, 1]$; threshold $\theta_{\mathrm{meta}} \in [0, 1]$. |
| $\kappa(e, u)$ | Element-level compatibility score in $[0, 1]$; threshold $\theta_{\mathrm{elem}} \in [0, 1]$. |
| $\mathrm{keys}(r)$ | Set of keys present in record $r$; $\mathrm{val}_k(r)$ is the value of $r$ at key $k$. |
| $\mathrm{age}(r)$ | Time elapsed since $r$ was created; $q_{\mathrm{src}}(r) \in [0, 1]$ source quality; $\#\mathrm{supports}(r)$ number of independent supporting references. |

---

**Algorithm 1** Adaptation Processing

---

**Input:** Experience $E$, Schema set $S$, Thresholds $\theta_{\mathrm{meta}}, \theta_{\mathrm{elem}}$
**Output:** Adaptation path and updated schema set
 1: Preprocess $E$ into a set of segments $\{e_1, e_2, ..., e_k\}$
 2: **for** each experience segment $e$ in $\{e_1, e_2, ..., e_k\}$ **do**
 3:     Find the best matching schema: $\sigma^* \leftarrow \arg\max_{\sigma \in S} s_{\mathrm{meta}}(e, \sigma)$
 4:     Let $s^* \leftarrow s_{\mathrm{meta}}(e, \sigma^*)$
 5:     **if** $s^* \geq \theta_{\mathrm{meta}}$ **then**
 6:         Find the best matching element: $u^* \leftarrow \arg\max_{u \in U(\sigma^*)} \kappa(e, u)$
 7:         Let $\kappa^* \leftarrow \kappa(e, u^*)$
 8:         **if** $\kappa^* \geq \theta_{\mathrm{elem}}$ **then**
 9:             SCHEMAUPDATE$(e, u^*)$ {Path: Assimilation/Update}
10:         **else**
11:             SCHEMAEVOLUTION$(e, \sigma^*)$ {Path: Accommodation/Evolve}
12:         **end if**
13:     **else**
14:         SCHEMACREATION$(e)$ {Path: Accommodation/Create}
15:     **end if**
16: **end for**
17: Apply Conflict Detection and Deactivation to all modified schemas in $S$[2]
18: **return** Adaptation results with paths and updated schema set $S$

---

**Update Operators.** **SchemaUpdate** incorporates a new record $r(e)$ into the matched element $u^*$:
$$V(u^*) \leftarrow V(u^*) \cup \{r(e)\}, \quad \text{s.t. } K_b \subseteq \mathrm{keys}(r(e)).$$
**Schema Evolution** creates a new element $u^{\mathrm{new}}$ with initial record:
$$U(\sigma^*) \leftarrow U(\sigma^*) \cup \{u^{\mathrm{new}}\}, \quad V(u^{\mathrm{new}}) \leftarrow \{r(e)\}.$$
**Schema Creation** instantiates a new schema $\sigma^{\mathrm{new}}$ with meta $m(e)$:
$$S(b) \leftarrow S(b) \cup \{\sigma^{\mathrm{new}}\}, \quad U(\sigma^{\mathrm{new}}) \leftarrow \emptyset.$$

---

[2]The detailed implementation of Conflict Resolution is described in Appendix C.2.

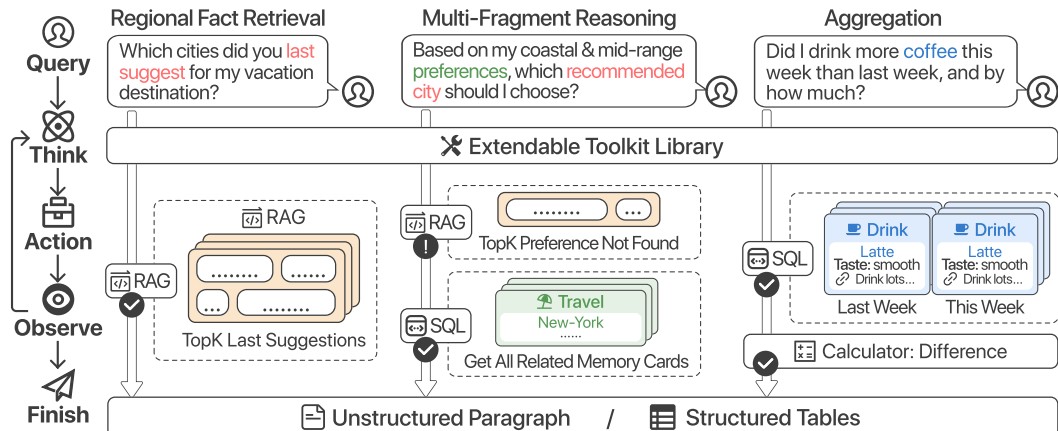

Figure 4: Retrieval Module of PISA. It supports diverse query handling through Regional Fact Retrieval, Multi-Fragment Reasoning, and Aggregation, leveraging an extendable toolkit library (*e.g.*, RAG, SQL, and Caculator) to utilize both unstructured and structured memory.

### 2.4 RETRIEVAL MODULE

The Retrieval Module is designed as a ReAct AI agent (Yao et al., 2023), built upon the Langgraph framework (LangChain Team, 2023). Its primary function is to interpret and analyze user queries and then retrieve pertinent information from the memory system. This process is intelligently guided by the inherent structure of the memories it accesses.

To adeptly handle a diverse range of user inquiries, the Retrieval Module is equipped with an extensible toolkit library, enabling it to select the most efficient tools based on the query's nature. As illustrated in the Figure 4, the module categorizes queries and deploys corresponding strategies:

- **For Regional Fact Retrieval Queries**, which are typically single-hop questions like *"Which cities did you last suggest for my vacation?"*, the AI agent employs a Retrieval-Augmented Generation (RAG) tool (Lewis et al., 2020). This allows for direct and rapid retrieval of specific memory fragments, such as past suggestions, from the memory pool.

- **For Multi-Fragment Reasoning Queries** that require synthesizing information from various sources, such as *"Based on my coastal & mid-range preferences, which recommended city should I choose?"*, the AI agent orchestrates a chain of tools. It might first use the RAG tool to pull up all related memory cards about travel recommendations and user preferences. Subsequently, it can leverage a SQL tool to perform structured queries across these memories to filter and reason, ultimately providing a tailored recommendation.

- **For Aggregation Queries** that necessitate calculation or data comparison, like *"Did I drink more coffee this week than last week, and by how much?"*, the module utilizes SQL to fetch relevant structured data (*e.g.*, coffee consumption logs for "This Week" and "Last Week"). Following the data retrieval, it can invoke a calculator tool to perform the necessary aggregation, such as calculating the difference, to deliver a precise answer.

## 3 EXPERIMENTS

### 3.1 DATASETS

To evaluate the performance of AI agent memory across different tasks, we primarily used two types of tasks to assess the memory system. The first is LOCOMO (Maharana et al.,

Table 1: Performance on LOCOMO benchmark. The evaluation metric is LLM-as-a-Judge accuracy. **Bold** and underline respectively highlight the best and second-best scores among external-memory methods; FullContext is listed for reference only.

| Method | Single-Hop | Multi-Hop | Temporal | Open-Domain | Average |
|---|---|---|---|---|---|
| Zep | 46.65 | 14.08 | 36.54 | 39.13 | 34.10 |
| MemoryOS | 43.06 | 19.01 | 16.67 | 47.83 | 38.40 |
| AMem | 56.94 | 28.87 | 71.79 | 50.00 | 51.90 |
| LangMem | 65.79 | 31.69 | 71.15 | 47.83 | 54.12 |
| Mem0 | 67.70 | 36.62 | 69.87 | 47.83 | 55.51 |
| PISA | **73.44** | **47.18** | **75.00** | **58.70** | **63.58** |
| FullContext | 78.71 | 49.30 | 87.18 | 58.70 | 68.47 |

2024b), which is used to evaluate long-term conversational memory. The second is AggQA, which is used to evaluate memory for data analysis tasks. LOCOMO contains 10 extended conversations. For a comprehensive assessment of long-term conversational memory, LO-COMO divides all questions into five distinct reasoning categories: (1) single-hop questions require answers based on a single session; (2) multi-hop questions require synthesizing information from multiple different sessions; (3) temporal reasoning questions can be answered through temporal reasoning and capturing time-related data cues within the conversation; (4) open-domain knowledge questions can be answered by integrating a speaker's provided information with external knowledge such as commonsense or world facts; (5) Adversarial questions are designed to trick the AI agent into providing wrong answers, with the expectation that the AI agent will correctly identify them as unanswerable.[3] AggQA is designed by ourselves, divided into medical and finance domains, focusing on data analysis tasks in these two fields. Questions are categorized into three difficulty levels: easy, medium, and hard, with the medical domain containing 49 questions and the financial domain containing 85 questions. More details about AggQA are presented in Appendix D.

### 3.2 BASELINES

AMem (Xu et al., 2025) proposes dynamic, self-evolving knowledge networks, built on atomic notes, autonomous linking, and continuous evolution for semantic understanding and efficient operation.

Mem0 (Chhikara et al., 2025) introduces a scalable memory architecture that dynamically manages salient information, ensuring consistency and improving reasoning efficiency.

LangMem (AI, 2025) tackles cross-session learning, personalization, and consistency via memory APIs, in-conversation tools, and background knowledge integration, with flexible storage compatibility and native LangGraph support.

Zep (Rasmussen et al., 2025) presents a temporal knowledge graph (Graphiti) synthesizing unstructured conversations and structured business data, achieving low-latency, scalable enterprise-grade memory.

MemoryOS (Kang et al., 2025) designs a hierarchical memory system (Short/Mid/Long-Term) with a heat-based update mechanism and two-tier retrieval for efficient management and access.

Full-Context directly providing the complete information as context to GPT-5 Mini (OpenAI, 2025) and instructing it to answer questions based on this context.

### 3.3 EVALUATION METRICS AND MAIN RESULTS

For the LOCOMO dataset and AggQA, we employ the LLM-as-a-Judge framework to evaluate the accuracy of responses. In the case of the LOCOMO dataset, we adopt a strict

---

[3]We excluded the adversarial category from our experiments.

Table 2: Performance on AggQA benchmark. The evaluation metric is LLM-as-a-Judge accuracy.

| Method | Med. | | | | Fin. | | | |
|--------|------|--------|------|---------|------|--------|------|---------|
| | Easy | Medium | Hard | Overall | Easy | Medium | Hard | Overall |
| MemoryOS | 18.18 | 12.00 | 0.00 | 10.20 | 17.39 | 14.89 | 7.14 | 14.29 |
| Zep | 9.09 | 40.00 | 0.00 | 22.45 | 8.70 | 10.42 | 7.14 | 9.41 |
| AMem | 27.27 | 28.00 | 0.00 | 20.41 | 39.13 | _25.00_ | _42.86_ | _30.59_ |
| Mem0 | 27.27 | 40.00 | _15.38_ | 30.61 | 39.13 | 16.67 | 21.43 | 23.53 |
| LangMem | _72.73_ | _64.00_ | 7.69 | _51.02_ | _43.48_ | 18.75 | 0.00 | 22.35 |
| PISA | **90.91** | **96.00** | **53.85** | **83.67** | **78.26** | **62.50** | **78.57** | **69.41** |
| FullContext | 45.46 | 68.00 | 23.08 | 51.02 | 95.65 | 62.50 | 57.14 | 70.59 |

evaluation criterion: for answers involving multiple items (*e.g.*, a query about all the books a character likes), any omission of valid results is scored as zero. Such a criterion places strong emphasis on the completeness of multi-item answers, thereby imposing higher demands on the memory system's ability to synthesize and preserve comprehensive information.

Table 1 shows the results of PISA compared with baselines on the LOCOMO dataset. Across all four evaluation modes (*i.e.*, Single-Hop, Multi-Hop, Open-Domain, Temporal) our system achieves state-of-the-art performance, indicating its robustness and adaptability to diverse reasoning requirements.

On our AggQA benchmark illustrated in Table 2, which emphasizes data analysis tasks, PISA achieves an even larger performance margin over baselines. By selectively highlighting task-oriented attributes and retaining critical analytical information, PISA delivers more efficient and accurate retrieval than generic memory systems. These findings underscore the effectiveness of treating memory as an active, task-oriented process rather than a passive repository.

## 3.4 HYPERPARAMETER ANALYSIS

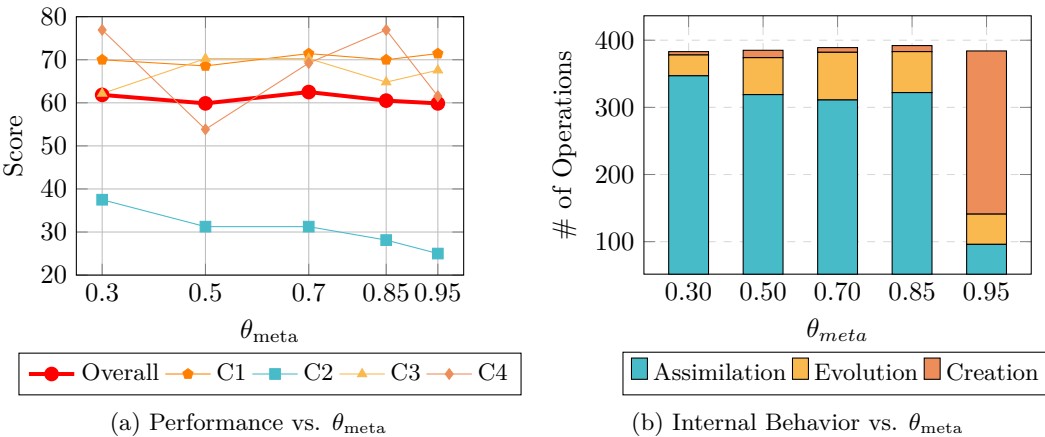

(a) Performance vs. $\theta_{\text{meta}}$    (b) Internal Behavior vs. $\theta_{\text{meta}}$

Figure 5: Analysis of the impact of $\theta_{\text{meta}}$ on agent performance and internal behavior. C1 to C4 represent Single-Hop, Multi-Hop, Temporal, and Open-Domain tasks, respectively.

We perform an hyperparameter analysis on conv-26 of the LOCOMO dataset to analyze the impact of the meta-cognition threshold, $\theta_{\text{meta}}$, on agent performance and internal behavior, as shown in Figure 5. This threshold governs the agent's decision to either assimilate new experiences into existing metas or create new ones.

Table 3: Detailed ablation studies on the AggQA benchmark, showing both Accuracy (Acc.) and Evidence Coverage (Cover.). PISA$^{-I}$ performs an ablation on the Initialization Module (on Med.), while PISA$^{-R}$ performs an ablation on the Retrieval Module (on Fin.).

| Method | Domain | Easy | | Medium | | Hard | | Overall | |
|---|---|---|---|---|---|---|---|---|---|
| | | Acc. | Cover. | Acc. | Cover. | Acc. | Cover. | Acc. | Cover. |
| PISA$^{-I}$ | Med. | 72.73 | 68.18 | 76.00 | 65.48 | 53.85 | 63.69 | 69.39 | 65.61 |
| PISA | | 90.91 | 83.27 | 96.00 | 86.60 | 53.85 | 82.77 | 83.67 | 84.84 |
| PISA$^{-R}$ | Fin. | 47.83 | 53.39 | 54.17 | 61.65 | 50.00 | 40.71 | 51.77 | 55.97 |
| PISA | | 78.26 | 83.13 | 62.50 | 75.02 | 78.57 | 65.93 | 69.41 | 75.72 |

As depicted in Figure 5a, the agent's overall performance is robust for $\theta_{\text{meta}}$ values ranging from 0.30 to 0.95. The peak score is achieved at 0.70, which demonstrates strong and balanced performance on most evaluation modes. Figure 5b illustrates the trade-off in memory operations. A lower $\theta_{\text{meta}}$ (*e.g.*, 0.30) encourages Assimilation, leading to frequent updates of existing memories. Conversely, a higher threshold (*e.g.*, 0.95) markedly increases Creation operations, resulting in a larger and more fragmented memory space. The value of $\theta_{\text{meta}} = 0.70$ strikes an effective balance, promoting a healthy mixture of Assimilation and Evolution without excessive Creation. Based on its high overall score and well-balanced internal mechanics, we set $\theta_{\text{meta}} = 0.70$ as the default value for our main experiments.

## 3.5 Ablation Study

We performed two targeted ablation studies on AggQA to validate our core design choices, with results shown in Table 3.

First, we tested the Initialization Module by removing its detailed, schema-driven setup on the Med dataset (PISA$^{-I}$). The resulting 14.28% drop in overall accuracy confirms that proactively building a task-oriented memory scaffold is critical for organizing information effectively and guiding downstream reasoning.

Second, we evaluated the Retrieval Module by 'blinding' it to the memory's hierarchical structure on the Fin dataset (PISA$^{-R}$). This led to a 17.64% decrease in overall performance, with a severe drop on hard tasks (from 78.57% to 50.00%). This demonstrates that structural awareness is vital for the agent to efficiently orchestrate tools and solve complex analytical queries.

## 4 Conclusions and Limitations

In this work, we introduced PISA, a structured memory framework grounded in Piaget's cognitive theory. Its unified schema architecture leverages task priors to build compact, query-efficient memory structures, complemented by scalable retrieval mechanisms. To address the narrow scope of existing benchmarks—often limited to factual text retrieval, we further designed AggQA, targeting structured data integration in medicine and finance. Empirical results show that PISA not only excels on the LOCOMO suite but also achieves strong performance on AggQA, highlighting its effectiveness for both general and task-oriented memory tasks.

A limitation of PISA lies in its reliance on prior knowledge. Moreover, our evaluation of task-oriented memory tasks has thus far been restricted to the AggQA benchmark we constructed. We aim to extend future testing to a broader range of benchmarks, which will allow for further optimization and validation of our approach.

## Reproducability Statement

All experiments were conducted using "GPT-5 Mini" as the core LLM with a default temperature of 1.0. For persistent storage of experimental data and metadata, we utilized PostgreSQL as the primary backend for data storage. For baseline comparisons, memory systems were configured as follows: **AMem** and **LangMem** use in-memory storage; **Full Context** uses no external memory; **Mem0** uses a local Qdrant instance; **Zep** uses the zep-cloud service; and **MemoryOS** uses a tiered memory system saved to local JSON files with in-memory FAISS for retrieval. The source code of PISA and AggQA dataset are available at https://anonymous.4open.science/r/PISA-421/.

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

CONTENTS

## A    LLM Usage Statement

In the preparation of this paper, the use of large language models (LLMs) was strictly confined to refining the final text. The authors first independently authored and completed full drafts of the introduction, related works, and PISA sections, ensuring all arguments and findings were our own. Only after this original writing phase did we employ a set of LLMs, namely Gemini 2.5 Pro(Google, 2025), Claude 4.0 Sonnet(Anthropic, 2025), and GPT-5(OpenAI, 2025), for the purpose of polish and refinement. Our interaction with these models focused on two areas: identifying and correcting potential grammatical or semantic errors, and gathering alternative phrasing suggestions to improve the manuscript's readability and coherence. The authors carefully reviewed all suggestions, and the final text is the result of our deliberate consideration and integration of these editorial recommendations. We affirm that the authors are solely and entirely responsible for the substance and accuracy of this work.

## B    Related Works

### B.1    Memory For LLM-based AI Agents

Extensive research has been devoted to memory systems for LLM-based AI agents. Du et al. (2025) proposes a unified framework that analyzes prior works along three complementary dimensions: *Taxonomy* (representation), *Management* (organization and updating), and *Utilization* (retrieval and application). We review related studies following this perspective.

For *memory representation*, existing works have explored multiple approaches to balance structure and flexibility. Hierarchical representations organize memory at different levels of abstraction, enabling selective access and forgetting (Zhong et al., 2024). Structured multi-attribute notes, incorporating contextual descriptions, keywords, and embeddings, construct semantic networks supporting interconnection across memory items (Xu et al., 2025). Hybrid representations that integrate natural language with graph-structured relations further provide interpretability and efficiency for dynamic storage (Chhikara et al., 2025).

In terms of *memory management*, researchers investigate diverse strategies to support memory evolution. Dynamic updating and forgetting mechanisms, inspired by the Ebbinghaus curve, maintain memory relevance and naturalness (Zhong et al., 2024). System-inspired architectures, such as segmented-paging designs migrated from operating systems, enable scalable organization in dialogue scenarios (Kang et al., 2025). Autonomous evolution strategies driven by LLMs generate new links and maintain balanced states without explicit rules, improving adaptability to long-term interactions (Xu et al., 2025). Moreover, consistency management approaches, including quadruple operations and conflict resolution, ensure coherent updates under high-concurrency requirements (Chhikara et al., 2025).

With respect to *memory utilization*, prior work has emphasized retrieval and application strategies that enhance effectiveness in reasoning and interaction. Hierarchical retrieval combined with heat-driven updating allows context-relevant access for personalized responses (Kang et al., 2025). Entity–relation dual-path retrieval methods jointly exploit symbolic and embedding-based representations to improve accuracy in complex queries (Chhikara et al., 2025). In addition, compression-based methods have been adopted to retain informativeness while reducing storage overhead, supporting scalability in extended interactions.

Despite these advances, most memory systems remain *task-agnostic*, overlooking the integration of agent goals as priors. This leads to two challenges: limited specialization for heterogeneous tasks, and significant computational and storage costs when striving for broad generality. These limitations point to the need for *task-oriented memory systems* that explicitly incorporate agent objectives into representation, management, and utilization, thereby capturing task-relevant information more efficiently and adaptively.

## B.2 Retrieval Module for Memory System

The retrieval module is a crucial component of the memory system, and its implementation depends on the design of the memory system's memory Taxonomy and Management dimensions. Regarding the retrieval module, we have an insight: memory systems and retrieval modules are complementary, and the structural nature of memory systems is directly proportional to the flexibility of retrieval modules. Existing research validates our perspective: The Mem0 system(Chhikara et al., 2025) demonstrates the synergistic design of structured memory representation and multi-granularity retrieval mechanisms, empirically proving that the structural transparency of memory systems directly determines the flexibility of retrieval modules. Its graph-based Mem0g architecture, with its high structural nature, supports more complex multi-hop and temporal query capabilities, validating the proportional relationship between structured memory representation and flexible retrieval performance. MemoryOS(Kang et al., 2025) constructs a highly structured memory system through its hierarchical storage architecture and heat-based dynamic update mechanisms. Its retrieval module relies on semantic segmentation and multi-level granularity strategies (such as segment-page two-level retrieval and personalized attribute integration) to achieve flexible and precise multi-dimensional memory access, confirming the complementary relationship between structured memory systems and retrieval flexibility. AMem(Xu et al., 2025) is a self-organizing and self-evolving memory system. Its memory Taxonomy dimension adopts dynamically generated multi-attribute structured notes, while the Management dimension introduces autonomous evolution mechanisms. Its unstructured memory storage leads to retrieval limitations, resulting in its ability to retrieve only through returning top-k notes. The reason for the complementary nature between memory systems and retrieval modules lies in the fact that the higher the degree of structuralization of the memory system, the more transparent its structure becomes, enabling the retrieval module to perform multi-granularity retrieval based on memory, thus achieving more flexible retrieval capabilities. Therefore, the design of retrieval modules should strictly consider the structure of memory systems and utilize the structural information of memory systems for retrieval as much as possible. Meanwhile, we propose our viewpoint: the design of memory systems, in addition to starting from the memory system itself, considering memory system implementation from the perspective of retrieval module design is also an effective way to think about memory system implementation. PISA's design not only starts from the memory system perspective but also from the retrieval module perspective, integrating both to achieve a memory system capable of multi-granularity information acquisition.

## B.3 LLM-based AI Agent Memory Benchmark

The evaluation of long-context capabilities in large language models (LLMs) has evolved from testing models' internal memory to specifically assessing external memory systems. Early research primarily focused on testing the limits of models' internal memory mechanisms. RULER (Hsieh et al., 2024) significantly extended the traditional "needle-in-a-haystack" test by constructing a synthetic benchmark with configurable length and complexity, introducing tasks such as multi-needle retrieval, multi-hop tracking, and aggregation. Although RULER provides a multi-faceted evaluation of models' parametric memory, its synthetic data design fails to reflect memory requirements in real-world application scenarios.

LOCOMO (Maharana et al., 2024a) constructed long-term dialogues incorporating event graphs and multimodal elements through a human-AI collaborative process, evaluating models' long-term memory capabilities in scenarios closer to real conversations. This work revealed the challenges LLMs face in understanding long-term temporal and causal dynamics, highlighting the need for a more comprehensive evaluation framework capable of handling long-term dependencies and complex causal relationships.

As research deepened, benchmarks specifically designed for external memory systems began to emerge. LongMemEval (Wu et al., 2025) assessed a memory assistant's multiple core long-term memory abilities (*e.g.*, extraction, reasoning, updating, and abstention) in multi-scenario ultra-long dialogues. MemoryAgentBench (Hu et al., 2025) clearly defined the four core capabilities of memory AI agents and, addressing the limitations of existing datasets

(*e.g.*, short context or one-time full-text input), innovatively proposed a unified evaluation framework with segmented input. However, as noted by research from the systems development frontline, the field still faces a challenge of benchmark scarcity: existing options often lack robustness and complexity, frequently limited to simple "needle-in-a-haystack" retrieval Rasmussen et al. (2025), and no benchmark can adequately evaluate a system's ability to integrate dialogue history with structured business data for comprehensive processing Rasmussen et al. (2025).

In terms of task complexity, benchmarks like `WikiTableQuestions` (Kweon et al., 2023) and `HybridQA` (Chen et al., 2020) advanced the development of hybrid reasoning over structured data and text. Subsequent research (*e.g.*, `HeteQA` (Yu et al., 2025)) further extended the complexity of cross-domain and multi-hop table QA. However, these works are generally built upon static, closed-domain data, with limited context length and static interaction patterns, primarily targeting single-query response scenarios, thus, it also fails to meet the requirements for continuous memory maintenance and complex numerical aggregation in dynamic dialogues.. Furthermore, existing literature shows insufficient discussion on scalability metrics crucial for production systems, such as cost and latency (Rasmussen et al., 2025).

In summary, the evaluation paradigm in this field has undergone a clear evolutionary path: from assessing models' internal memory (RULER, LOCOMO), to focusing on the systematic evaluation of long-term memory (LongMemEval), and then to designing comprehensive benchmarks specifically for external memory AI agents (MemAgentBench). However, significant gaps remain in existing benchmarks: they either lack investigation into professional domain-specific discrete data and numerical aggregation capabilities, or struggle to evaluate memory mechanisms under long dialogues, and none systematically address the integrated processing of dialogue history and business table data. To this end, we propose AggQA, which discretely embeds tabular data within 30K-token-level professional dialogues, aiming to systematically evaluate the numerical perception, reconstruction, and aggregation capabilities for external memory systems, thereby filling the gaps in the current evaluation system.

## C    Additional Material of PISA

### C.1    Piaget's Constructivist Theory: A Detailed Computational Analogy

The design philosophy of the PISA memory system is rooted in the constructivist theory of Piaget et al. (1952), which provides a profound conceptual blueprint for understanding and building intelligent agents capable of autonomous learning. This section aims to dissect the core concepts of this theory in detail and clarify how they are translated into the specific computational mechanisms of the PISA system.

#### C.1.1    Schema: The Structural Unit of Knowledge

In Piaget's schema theory, a "Schema" is the basic structural unit of cognition, a mental representation or knowledge framework that an individual uses to understand and respond to the environment (Piaget et al., 1952). It is not static information but a dynamic, organized program, such as an infant interacting with objects through a "grasping" schema.

In PISA, this concept is directly mapped to a structured unit of knowledge within the system, also called a "Schema." Each PISA schema is a computational object that encapsulates knowledge for a specific task, stored in the `Schema Engine`. It contains not only descriptive knowledge (*e.g.*, task goals, application conditions) but, more importantly, procedural knowledge (*e.g.*, a sequence of actions or API calls required to complete the task). This makes PISA's schemas consistent with Piaget's definition: they are executable "mini-programs" for interacting with the world.

### C.1.2 Assimilation: Integrating the Familiar

"Assimilation" is the process of integrating new information into existing schemas when an individual encounters it. For example, a child who has formed a "dog" schema will classify a new, different breed of dog into the "dog" category, thereby strengthening and generalizing the existing schema. Assimilation is an efficient process of understanding the world using existing knowledge.

PISA simulates "assimilation" through the `Retrieval Module` and subsequent successful execution. When a new task arrives:

1. The `Retrieval Module` first attempts to match the task with an existing schema in the `Schema Engine`.

2. If a matched schema is successfully executed (*i.e.*, the agent completes the task smoothly), this constitutes an "assimilation" event.

3. As a result, the confidence or weight of that schema is increased, making it more likely to be retrieved in the future.

This process is illustrated in Figure 6, which shows how new experiences are integrated into existing cognitive frameworks. The initial "cat" schema includes attributes such as color (brown), legs (4), ears (2), and tail (1). When a "new experience" with new color information (a slightly darker brown) is encountered, the system readily integrates the new color into the existing "cat" schema, forming an updated "assimilated cat" schema where the color attribute is updated while other basic attributes remain unchanged. This ensures that the system can efficiently utilize and consolidate validated successful experiences, achieving stable application of knowledge.

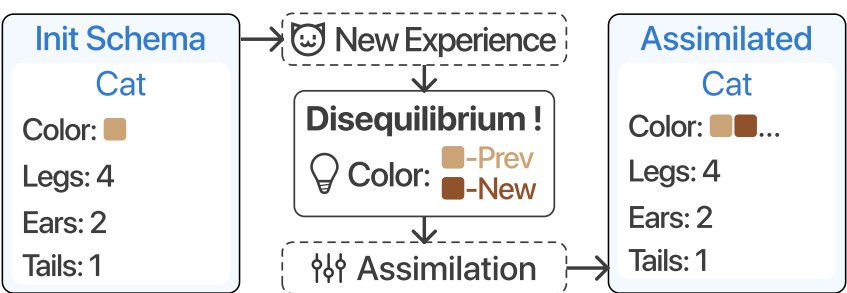

Figure 6: Assimilation: Integrating new experiences into existing cognitive frameworks.

### C.1.3 Accommodation: Adapting to the Novel

"Accommodation" occurs when existing schemas cannot explain new information or lead to failure, a state known as cognitive disequilibrium. At this point, the cognitive system must adjust old schemas or create new ones to adapt to the new challenge. For example, when a child sees a cat for the first time, trying to understand it with the "dog" schema will create a conflict (*e.g.*, different sounds), forcing the child to adjust their cognitive structure and eventually form a new "cat" schema.

PISA computationalizes the "accommodation" process and uses it as the core mechanism for dealing with failures and novel environments, primarily handled by the `Adaptation Module` (its position in the overall architecture is shown in Figure 2). When a task execution fails or an unexpected result occurs:

1. The system identifies this as a "Cognitive Disequilibrium," indicating a flaw in the current schema.

2. The `Adaptation Module` is activated, and it analyzes the cause of the failure (*e.g.*, incorrect API parameters, incomplete action sequence).

3. Based on the analysis, the module performs a structural modification on the failed schema (*e.g.*, correcting parameters, adding/deleting steps) or creates a completely new schema to solve the problem.

This mechanism is implemented in two ways, as shown in Figure 7 and Figure 8.

- Figure 7 depicts the "Accommodation (modify)" process: When the initial schema for "coffee" (Taste: Bitter, Color: Brown, Effect: N/A, Smell: N/A) encounters an "actual experience" and creates "disequilibrium" (Effect: Refreshing, Smell: Aromatic), the existing schema is insufficient to fully explain the new information. Through the "accommodation" process, the system modifies the original "coffee" schema, adding the "Refreshing" effect and "Aromatic" smell attributes, forming a more complete "accommodated coffee" schema.

- Figure 8 depicts the "Accommodation (create)" process: When the initial schema for "dog" (Sound: Woof) encounters an "actual experience" and creates "disequilibrium" (Sound: Meow), the existing "dog" schema cannot handle the novel "Meow" sound. Through the "accommodation" process, the system creates a brand new "cat" schema that includes the "Meow" sound attribute, thereby effectively handling the new information.

This mechanism endows PISA with true learning capabilities, enabling it to extract knowledge from failure and achieve fundamental growth in its cognitive structure.

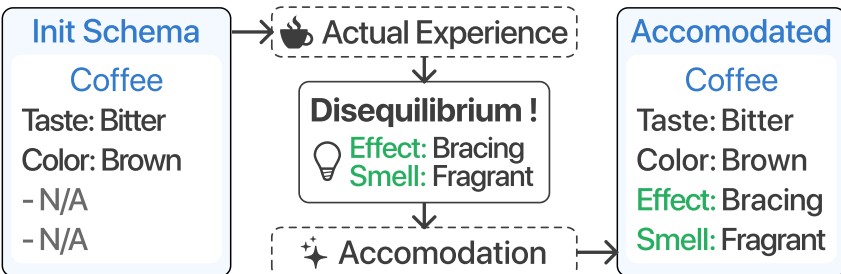

Figure 7: Accommodation (modify): Modifying existing schemas when they fail to adequately account for novel information.

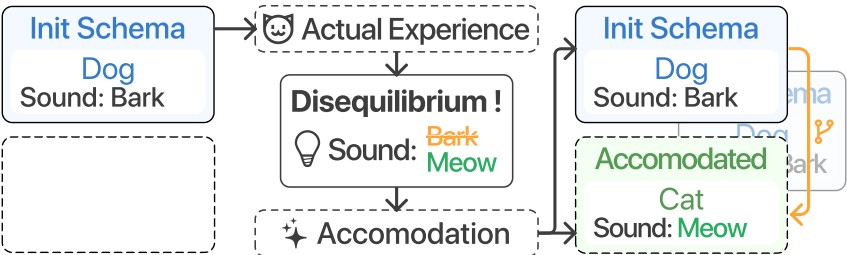

Figure 8: Accommodation (create): Creating new schemas when existing structures cannot handle novel information.

### C.1.4 Equilibration: The Drive for Balance

"Equilibration" is the internal driving force behind all cognitive development (Piaget et al., 1952). It is the process of seeking a balance between assimilation and accommodation. The system strives to achieve a stable state of equilibrium, but new challenges constantly disrupt this balance, driving the system to achieve a higher, more adaptive new equilibrium through accommodation. Cognitive development thus emerges from this dynamic equilibrium between assimilation and accommodation.

In PISA, "equilibration" is manifested in the system's continuous pursuit of an efficient and accurate schema library. The system defaults to efficient "assimilation," but when "cognitive disequilibrium" (task failure) accumulates, it will activate "accommodation" at a higher cost to correct the model, aiming to reach a new, more powerful "equilibrium." This dynamic balancing mechanism allows PISA to resolve the fundamental "stability-plasticity" dilemma, building a memory system that is both stable in operation and capable of continuous evolution.

## C.2 Conflict Resolution in Memory

To maintain the integrity and consistency of the stored information, PISA incorporates a sophisticated conflict resolution mechanism. This process involves detecting conflicting records, evaluating their reliability, and deactivating those that are less reliable.

### C.2.1 Conflict Detection

Within each memory element $u$, which contains a set of records $V(u)$, conflicts between any two records $r_i$ and $r_j$ are identified using the following contradiction function:

$$\text{Contr}(r_i, r_j) = \mathbb{I}[\exists k \in K_b \cap \text{keys}(r_i) \cap \text{keys}(r_j) : \text{conflict}(\text{val}_k(r_i), \text{val}_k(r_j))].$$

Here, $K_b$ represents the set of keys for a given bucket $b$. A conflict is flagged if any shared key $k$ has conflicting values. The conflict$(\cdot, \cdot)$ predicate is domain-specific; for instance, it could check for inequality of canonical strings for categorical keys or a value difference exceeding a tolerance for numerical keys.

### C.2.2 Record Reliability Scoring

Each record $r$ is assigned a reliability score $s(r)$ based on three factors: recency, source quality, and community support. The score is calculated as:

$$s(r) = w_{\text{recency}} \cdot (1 + \text{age}(r))^{-1} + w_{\text{source}} \cdot q_{\text{src}}(r) + w_{\text{support}} \cdot \#\text{supports}(r).$$

where $\text{age}(r)$ is the age of the record, $q_{\text{src}}(r)$ is the quality score of its source, and $\#\text{supports}(r)$ is the number of other records that support it. The weights $w_{\text{recency}}$, $w_{\text{source}}$, and $w_{\text{support}}$ are nonnegative weights that sum to 1, controlling the relative importance of these factors.

### C.2.3 Conflict Deactivation

Conflicts are resolved by deactivating less reliable records. We construct a conflict graph $G_u$ for each element $u$, where an edge exists between any pair of records with Contr = 1. For each connected component $C$ in this graph, a single "winner" record is chosen to remain active, while all others in the component are deactivated. The active status $\alpha(r)$ is determined by:

$$\alpha(r) = \begin{cases} 1, & \text{if } r = \underset{r' \in C}{\arg\max}\, s(r') \\ 0, & \text{otherwise.} \end{cases}$$

Ties are broken deterministically, first by preferring the record with the smaller $\text{age}(r')$, and then by the larger source quality score $q_{\text{src}}(r')$. This winner-takes-all approach ensures that only the most reliable and consistent information is retained.

## C.3 MCP Tools of Retrieval Module

In the retrieval module of PISA, we implement a set of tools that adhere to Anthropic's Model-Context Protocol (MCP)(Anthropic, 2025). These tools operate as independent back-end microservices and communicate with the upstream agent via standard input/output (STDIO). This design offers several core advantages. First, functional decoupling allows each tool to be developed, tested, and deployed independently, enhancing the system's maintainability. Second, running tools as separate processes provides a foundation for fine-grained security control and resource isolation, ensuring system stability. Finally, adherence

**RAG Service**   The RAG Service is responsible for retrieving unstructured and semi-structured data. Built upon a vector database, it performs semantic similarity searches based on natural language queries. To ensure data freshness, we have designed a periodic update mechanism that incrementally incorporates the latest information into the index. This service is crucial for PISA's ability to comprehend extensive conversation histories, external documents, and knowledge bases.

**Structural Query Service**   To facilitate secure access to structured data, we have implemented an SQL Service. This service acts as a proxy layer to the database. It receives SQL query requests, which are often generated from natural language, and subjects them to rigorous syntax checks and security audits before execution to prevent malicious attacks such as SQL injection. All queries are executed within a restricted sandbox environment, exposing only necessary database views, thereby protecting the integrity of the underlying data.

**Calculator Service**   To empower the agent with the ability to perform precise calculations, we provide a Calculator Service. This service executes mathematical expressions within an isolated sandbox environment, supporting basic arithmetic operations, variable assignments, and function calls. This method of isolated execution mitigates the risks associated with unsafe operations like 'eval()', ensuring the security of computational tasks.

**Collaborative Workflow**   These tools do not operate in isolation but are orchestrated by the agent to accomplish complex tasks collaboratively. A typical workflow is as follows: The agent first retrieves relevant information from the knowledge base using the RAG Service, then queries for precise data from a structured database using the SQL Service, and finally may use the Calculator Service to perform statistical analysis on the query results. The entire process communicates via the MCP protocol, ensuring standardized data exchange and procedural reliability. Through this approach, PISA's retrieval module can efficiently and securely integrate information from diverse sources to provide decision support for upstream applications.

# D   AGGQA

## D.1   DATASET CONSTRUCTION

Figure 9 illustrates the comprehensive pipeline for constructing the AggQA benchmark. The process is organized into four main stages: *Corpus Collection*, where source data is gathered and preprocessed; two parallel generation tracks named *QA-Pair Generation* and *Session Simulation*; and a final *Manual Refinement* stage to ensure data quality. In the parallel tracks, we generate structured question-answer items and context-rich dialogues independently. Both streams of data then converge into the unified human verification stage, ensuring all benchmark components meet our quality standards.

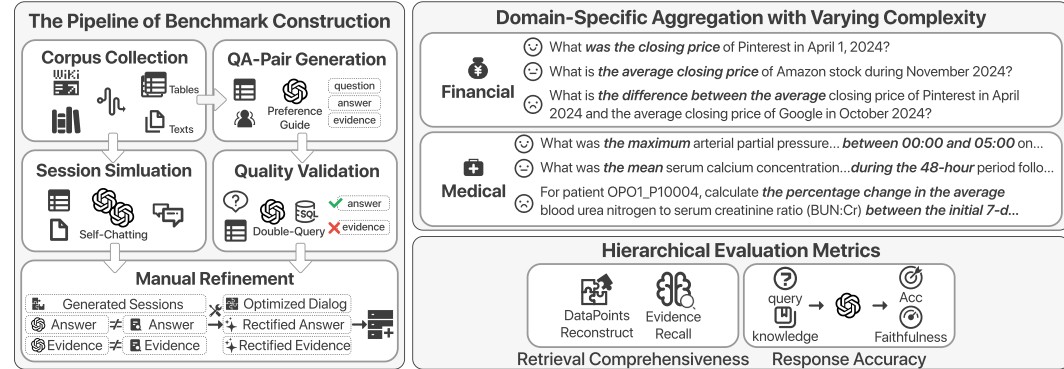

Figure 9: The construction pipeline of the AggQA benchmark. The process flows from Corpus Collection to two parallel generation tracks (QA-Pair Generation and Session Simulation), and culminates in a unified Manual Refinement stage.

### D.1.1 SOURCE CORPORA AND PREPROCESSING

The foundation of AggQA lies in two distinct, high-quality datasets, which we preprocess to extract and structure the numerical data essential for our tasks. Table 4 provides an overview.

| Domain | Source Dataset | Data Type | License |
|--------|---------------|-----------|---------|
| Financial | BizFinBench (Lu et al., 2025) | Textual Tabular | CC BY-NC 4.0 |
| Medical | PubMedQA (Jin et al., 2019) | Structured Abstracts | MIT License |

Table 4: Source Corpora Overview

**Financial Domain.** We utilize the `BizFinBench` dataset (Lu et al., 2025), focusing on its "Financial Numerical Computation" task. In this task, the required tabular data is often presented in a textual format directly within the question prompt. Our preprocessing involves parsing these prompts to programmatically extract and translate the textualized tables into structured English formats, while preserving professional terminology with standardized abbreviations. This process yields clean, well-structured financial tables that serve as the evidentiary basis for our generation tasks.

**Medical Domain.** We leverage the `PubMedQA` dataset (Jin et al., 2019). The source data is provided as JSON objects, where each entry contains a structured abstract with content segmented into labeled sections. Although the raw data has little tabular data, we developed a synthesis procedure that first parses these structured abstracts to build a rich vocabulary of specialized medical terminologies. This vocabulary is then used to generate synthetic yet realistic medical measurement records in a tabular CSV format.

### D.1.2 QA-PAIR GENERATION

This track is dedicated to producing aggregation-oriented, single-turn question-answering items that can be programmatically verified. It follows a dual-stream methodology.

The first stream is *LLM-driven*. An LLM is prompted with a snippet of continuous data from a preprocessed table and is tasked with generating a complete tuple: a question (`Q`), answer (`A1`), evidence (`E1`), an answer query (`SQL_Answer`), and an evidence query (`SQL_Evidence`). We execute these SQL queries to obtain the ground-truth SQL answer (`A2`) and SQL evidence (`E2`). The generated `A1`/`E1` are then matched against `A2`/`E2`, with any discrepancies flagged for human review.

The second stream is *script-driven.* For data exhibiting clear patterns, we employ Python scripts to generate precise answer-evidence pairs and a logical "hint". This structured information is then passed to an LLM, which focuses on formulating diverse, natural-language questions based on the pre-verified facts.

### D.1.3 Session Simulation

In this parallel track, we adapt the self-chat methodology (Xu et al., 2023) to generate multi-turn, evidence-coverage-oriented professional sessions. The process involves an LLM simulating a conversation between a User and an Assistant, where numerical data points from the source tables are discretely embedded into the dialogue. This technique creates a mixed information stream that intertwines structured data with conversational context, closely mimicking a real-world analytical task. To foster interaction diversity, the simulation operates in two distinct modes.

**Mode A: User-Provided Data.** The simulation starts with the User presenting a subset of data and asking for analysis, such as identifying a trend or explaining an anomaly. The Assistant responds directly based on the information provided by the User.

**Mode B: User-Sought Data.** The User initiates the conversation by seeking specific information. The Assistant, with access to the entire data table as its knowledge base, retrieves and presents the relevant data, often prompting further follow-up questions from the User.

### D.1.4 Manual Refinement

All instances from both the *QA-Pair Generation* and *Session Simulation* tracks converge at this final stage for human review. The main objective here is to validate the factual accuracy of answers and ensure a logical consistency between the dialogue, the question, and the supporting evidence. Our domain-expert annotators focus on confirming that the final answer is correctly derivable from the provided evidence and that the dialogue context sufficiently covers the necessary data points. This process filters out instances with clear factual errors or logical inconsistencies, thereby ensuring the overall reliability of the benchmark. Figures 10, 11, and 12 provide examples of instances that have passed or failed this refinement process.

### D.2 Dataset Analysis

To contextualize our contribution, we provide a detailed analysis of the AggQA benchmark. We first compare its structural and task-oriented characteristics with several established datasets and then delve into the internal distributions of the data itself.

### D.2.1 Basic Statistics and Comparison

Current evaluation benchmarks generally fall into two distinct categories. On one hand, long-context benchmarks like RULER (Hsieh et al., 2024), LoCoMo (Maharana et al., 2024a), and LongMemEval (Wu et al., 2025) assess memory over long, unstructured texts but typically focus on retrieving isolated facts. On the other hand, Table QA benchmarks such as FinQA (Chen et al., 2021) and HeteQA (Yu et al., 2025) test complex reasoning but always provide a complete, structured table upfront.

AggQA is designed to bridge this gap. It uniquely evaluates a system's ability to first reconstruct a latent table from numerical data discretely embedded within a long dialogue, and then aggregate this data. Table 5 situates AggQA relative to these existing benchmark families.

As the comparison illustrates, AggQA introduces a fundamentally new evaluation paradigm. Unlike long-context benchmarks, it moves beyond simple fact retrieval to require reasoning over interrelated numerical data. Unlike all listed table-based benchmarks, AggQA does not provide an explicit, clean table. Instead, its *Implicit & Discretized* data format forces

✅ **Financial: Session Simulate**

| code | sname | tdate | close |
|------|-------|-------|-------|
| NFLX.O | Netflix | 2024-06-03 | 0.507032 |
| NFLX.O | Netflix | 2024-06-04 | 0.505296 |
| | | . . . | |
| NFLX.O | Netflix | 2024-07-30 | 0.498064 |
| NFLX.O | Netflix | 2024-07-31 | 0.502680 |

User LLM:
"I've been tracking NFLX.O's closing prices in June–July 2024. Here's the trend:

- June 3: RMB 0.507 million
- June 10: RMB 0.516 million
- June 17: RMB 0.541 million
- June 24: RMB 0.535 million
- July 1: RMB 0.539 million
- July 8: RMB 0.549 million
- July 15: RMB 0.525 million
- July 22: RMB 0.518 million
- July 29: RMB 0.502 million

What could explain the sharp decline after mid-July and the earlier volatility?"

Assistant LLM:
"The sharp decline in NFLX.O's closing price after mid-July 2024, along with the earlier volatility, could be attributed to several factors, both company-specific and broader market-related. Here are some potential explanations:

1. Earnings Report or Guidance Disappointment

- Netflix (NFLX) may have released its Q2 2024 earnings in mid-July, and if the results missed expectations (*e.g.*, subscriber growth slowdown, weaker revenue, or lower-than-expected guidance), the stock could have sold off sharply.
- Investors might have been concerned about rising competition in streaming or slowing international growth.

2. . . .

Key Takeaway:
Without additional context (*e.g.*, USD price movements, earnings dates, or news headlines), the RMB-denominated decline likely stems from a combination of NFLX-specific weakness and RMB/USD forex dynamics. Checking Netflix's actual earnings date and analyst reports from that period would provide clearer answers."

Figure 10: An example of a correct self-chat session.

**Financial: Session Simulate**

| code | sname | tdate | net_flow | outflow |
|------|-------|-------|----------|---------|
| 300033.SZ | Tonghuashun | 2023-12-01 | 27900.0 | 50400.0 |
| 300033.SZ | Tonghuashun | 2023-12-04 | 57000.0 | 68000.0 |
| | | . . . | | |
| 300033.SZ | Tonghuashun | 2023-12-28 | 29400.0 | 63099.99 |
| 300033.SZ | Tonghuashun | 2023-12-29 | 9625.55 | 38600.0 |

User LLM: I've been looking at Tonghuashun's capital flow data for December 2023, particularly the mid-December period. For example, on Dec 11th there was a massive outflow of RMB 98,900 million but a positive net_flow of RMB 35,700 million, while just two days later Dec 13th saw a net outflow of RMB 14,858 million despite a lower outflow value. What factors might explain these rapid shifts in net flows relative to outflows during this period?"

Assistant LLM: The provided data *does not include entries for December 11th or December 13th*, which were referenced in your example. . . .
Error Reason: The assistant LLM failed to respond to the user's request directly because it treated remaining evidence as user-supplied context, thereby conflating privileged information with user input.

Figure 11: An example of a wrong self-chat session

Table 5: Descriptive comparison of AggQA. Its primary innovation is the "Discretized Table" format, which necessitates the novel challenge of "Reconstruction & Aggregation".

| Dataset | Context Modality | Numerical Data Format | Primary Challenge | Context (∼K token) |
|---------|-----------------|----------------------|-------------------|--------------------|
| **AggQA (Ours)** | Dialogue | Discretized Table | Reconstruction & Aggregation | ∼**56** |
| RULER (Hsieh et al., 2024) | Static Document | Isolated Facts | Fact Retrieval | $3 \sim 200$ |
| LoCoMo (Maharana et al., 2024a) | Dialogue | N/A | Causal Understanding | ∼9 |
| LongMemEval (Wu et al., 2025) | Dialogue | N/A | Memory Abilities | ∼115 |
| MemoryAgentBench (Hu et al., 2025) | Dialogue | N/A | Agentic Memory Use | ∼64 |
| HeteQA (Yu et al., 2025) | Static Document | Explicit Table | Multi-hop Table QA | - |
| HybridQA (Chen et al., 2020) | Static Document | Explicit Table | Hybrid QA | <16 |
| FinQA (Chen et al., 2021) | Static Document | Explicit Table | Numerical Reasoning | <4 |
| MedQA (Jin et al., 2021) | Static Document | N/A | Reading Comprehension | <2 |

**✖ Financial: Quality Validation**

| code | sname | tdate | close |
|------|-------|-------|-------|
| PINS.N | Pinterest | 2024-04-01 | 0.028256 |
| PINS.N | Pinterest | 2024-04-02 | 0.028104 |
| | | ... | |
| PINS.N | Pinterest | 2024-05-30 | 0.033104 |
| PINS.N | Pinterest | 2024-05-31 | 0.033192 |

Question: What was the average closing price of Pinterest stock from April 1 to April 30, 2024?

LLM answer with evidence: 0.0275488

| date | close | date | close | date | close | date | close |
|------|-------|------|-------|------|-------|------|-------|
| 04-01 | 0.028256 | 04-08 | 0.027184 | 04-15 | 0.026064 | 04-22 | 0.026184 |
| 04-02 | 0.028104 | 04-09 | 0.027728 | 04-16 | 0.026160 | 04-23 | 0.026416 |
| 04-03 | 0.028224 | 04-10 | 0.026800 | 04-17 | 0.026216 | 04-24 | 0.026280 |
| 04-04 | 0.027736 | 04-11 | 0.027216 | 04-18 | 0.026328 | 04-25 | 0.026112 |
| 04-05 | 0.027424 | 04-12 | 0.026832 | 04-19 | 0.025920 | 04-26 | 0.027168 |
| | | | | 04-29 | 0.027024 | 04-30 | 0.026760 |

SQL Answer Query:
```
SELECT AVG("close") FROM Table_close WHERE "code" = 'PINS.N' AND
"tdate" BETWEEN '2024-04-01' AND '2024-04-30'
```
SQL Evidence Query:
```
SELECT * FROM Table_close WHERE "code" = 'PINS.N' AND "tdate"
BETWEEN '2024-04-01' AND '2024-04-30'
```
SQL Answer: 0.02691527272727273
SQL Evidence:

| date | close | date | close | date | close | date | close |
|------|-------|------|-------|------|-------|------|-------|
| 04-01 | 0.028256 | 04-08 | 0.027184 | 04-15 | 0.026064 | 04-22 | 0.026184 |
| 04-02 | 0.028104 | 04-09 | 0.027728 | 04-16 | 0.026160 | 04-23 | 0.026416 |
| 04-03 | 0.028224 | 04-10 | 0.026800 | 04-17 | 0.026216 | 04-24 | 0.026280 |
| 04-04 | 0.027736 | 04-11 | 0.027216 | 04-18 | 0.026328 | 04-25 | 0.026112 |
| 04-05 | 0.027424 | 04-12 | 0.026832 | 04-19 | 0.025920 | 04-26 | 0.027168 |
| | | | | 04-29 | 0.027024 | 04-30 | 0.026760 |

Error Category: answer not match
Error Reason: wrong LLM calculation
Ground Truth: 0.02691527272727273

Figure 12: An example of a mismatch question-answer pair

models to actively search, identify, and mentally reconstruct the tabular structure from scattered mentions within a long dialogue. This dual challenge of *Numerical Reconstruction & Aggregation* represents a more realistic and complex test for memory systems, simulating how an agent must often synthesize structured insights from unstructured, conversational data streams.

### D.2.2 QUESTION AND EVIDENCE COMPOSITION

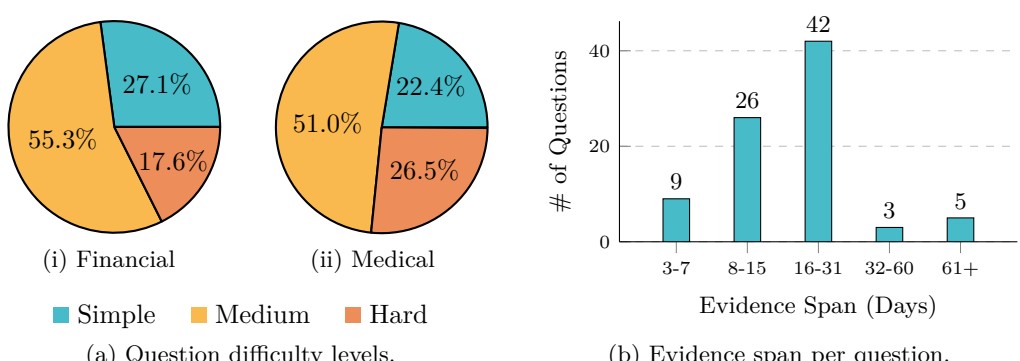

(i) Financial   (ii) Medical

■ Simple   ■ Medium   ■ Hard

(a) Question difficulty levels.   (b) Evidence span per question.

Figure 13: Composition of the AggQA benchmark. (a) shows the distribution of question difficulty, and (b) illustrates the evidence span required per question.

The AggQA benchmark is composed of 85 questions in the Financial domain and 49 in the Medical domain. To ensure a rigorous evaluation, the majority of these are designed to be of medium to high complexity. As illustrated in Figure 13a, 55.3% of Financial questions are "Medium" and 17.6% are "Hard". The Medical domain follows a similar pattern, with 51.0% "Medium" and 26.5% "Hard" questions, ensuring the benchmark robustly tests advanced reasoning capabilities.

A key metric for task complexity is the *evidence span*. For time-based questions, we define this as the duration in days of the relevant period, while for non-temporal questions, it is the number of discrete evidence items. As shown in Figure 13b, the distribution highlights a focus on medium- to long-term analysis. The vast majority of questions—nearly 80%—cover a span of between 8 and 31 days (one to four weeks). This design forces models to process and reason over extended sequences of data, rather than isolated facts, to derive an answer.

### D.2.3 SQL COMPLEXITY ANALYSIS

The ground-truth SQL queries provide a semantic blueprint of the required reasoning process for all 49 Medical questions and 41 of the 85 Financial questions. Our analysis of these queries, summarized in Table 6, reveals the profound logical and mathematical complexity embedded in the benchmark.

Table 6: SQL Complexity Analysis: A side-by-side comparison of component occurrence rates (% of questions with SQL) across domains.

| SQL Component | Financial (%) | Medical (%) |
| --- | --- | --- |
| Aggregate Functions (`AVG`, `SUM`, etc.) | 100.0 | 71.4 |
| Subqueries (Nested `SELECT`) | 68.3 | 34.7 |
| Arithmetic Operations (`+`, `-`, `*`, `/`) | 100.0 | 100.0 |
| Multiple `WHERE` Conditions | 100.0 | 100.0 |
| `GROUP BY` Clause | 9.8 | 10.2 |

As shown in the table, tasks in both domains consistently require complex logic, with 100% of SQL-based questions involving arithmetic operations and multiple filtering conditions. However, distinct patterns emerge. The Financial tasks are characterized by a higher dependency on multi-step reasoning, evidenced by the fact that 68.3% of its queries involve subqueries—nearly double the rate in the Medical domain (34.7%). Furthermore, every Financial SQL query utilizes an aggregate function. This high prevalence of complex logical structures ensures that AggQA robustly evaluates a model's capacity for sophisticated, multi-step numerical reasoning.

## D.3 DETAILED RESULTS

While the end-to-end accuracy presented in Table 2 provides a summary of overall performance, a deeper diagnosis is required to understand the specific failure points of each model. To this end, we introduce a critical intermediate metric: *Evidence Coverage.* We define this as the model's ability to successfully retrieve all discrete data points required to compute the final answer from the context. A high evidence coverage indicates strong data recall, while a low coverage points to failures in the initial retrieval stage.

Table 7 provides a side-by-side comparison of the overall accuracy with a detailed breakdown of evidence coverage across different difficulty levels. This analysis allows us to disentangle retrieval failures from reasoning failures, revealing whether a model's poor performance stems from an inability to find the correct information or an inability to compute with it correctly.

Table 7: Detailed comparison of Overall Accuracy (%) and Evidence Coverage (%) across difficulty levels for MED & FIN datasets.

| Method | Med. | | | | Fin. | | | |
|---|---|---|---|---|---|---|---|---|
| | Overall Acc. | Evidence Coverage Detail | | | Overall Acc. | Evidence Coverage Detail | | |
| | | Easy | Medium | Hard | | Easy | Medium | Hard |
| MemoryOS | 10.20 | 18.18 | 10.00 | 0 | 14.29 | - | - | - |
| Zep | 22.45 | 24.54 | 48.40 | 3.07 | 9.41 | 25.26 | 26.67 | 40.00 |
| AMem | 20.41 | 36.27 | 29.92 | 3.85 | 31.77 | 50.04 | 40.63 | 42.29 |
| Mem0 | 30.61 | 35.45 | 48.40 | 46.54 | 23.53 | 60.35 | 41.77 | 50.00 |
| LangMem | 51.02 | 50.00 | 54.72 | 43.23 | 22.35 | 44.22 | 23.63 | 0 |
| PISA | 69.39 | 68.18 | 65.48 | 63.69 | 51.77 | 53.39 | 61.65 | 40.71 |
| PISA* | **83.67** | **83.27** | **86.60** | **82.77** | **69.41** | **83.13** | **75.02** | **65.93** |
| FullContext | 51.02 | 50 | 54.72 | 43.23 | 70.59 | 97.09 | 84.04 | 53.71 |

