# OpenReview forum: "PISA: A Pragmatic Psych-Inspired Unified Memory System for Enhanced AI Agency"
_ICLR.cc/2026/Conference — ICLR 2026 Conference Withdrawn Submission_

### Official Review · Reviewer_sCCh · 2025-10-29

**Soundness:** 2
**Presentation:** 3
**Contribution:** 1
**Rating:** 2
**Confidence:** 3

**Summary:**

This paper proposes a new memory system designed to enhance AI agents in long-term conversational and data analysis tasks. Drawing inspiration from Piaget's cognitive development theory, PISA introduces a three-module architecture: an Initialization Module that constructs task-oriented memory structures through schema creation, an Adaptation Module that implements a tri-modal mechanism (assimilation, accommodation via schema evolution, and accommodation via schema creation) for updating memories, and a Retrieval Module designed as a ReAct AI agent (Yao et al., 2023). The memory is organized hierarchically into schemas containing meta (category), element (specific instance), element-value (attributes), and experience ID (linking to original conversation). The proposed method is evaluated on the LOCOMO benchmark and a newly introduced AggQA dataset using an LLM-as-a-judge metric, demonstrating performance improvements over current state-of-the-art systems.

**Strengths:**

**Novel memory architecture**: The authors introduce a psychologically-inspired approach to memory storage for AI agents that exploits hierarchical schema structures to create, access, and update memories more effectively.

**Comprehensive system description**: The paper provides detailed explanations of each architectural component and presents algorithms necessary for system functioning, with a commitment to make code publicly available.

**New evaluation benchmark**: The authors introduce AggQA dataset, to evaluate models on data analysis tasks across medical and finance domains with varying difficulty levels.

**Weaknesses:**

**Presentation clarity**: Section 2 suffers from verbose, nested explanations that would benefit from more formal mathematical notation. Concepts introduced in the "Notation and Symbols" table (line 220) are insufficiently explained in the main text and Figure 3, such as multiple elements belonging to the same schema.

**Unclear attribution of contributions**: For the Retrieval Module (Section 2.4 and Figure 4), the distinction between components taken from Yao et al. (2023) and those newly introduced or adapted for PISA is not clearly delineated.

**Ad-hoc design concerns**: Throughout the paper, extensive notation is introduced (Memory Pool, Buckets, Schema, Element, etc.) without clarifying whether these are created specifically for this work or inspired by existing literature. The categorization of queries in Section 2.4 appears constructed to target specific task types found in the evaluation datasets (e.g., Regional Fact Retrieval -> LOCOMO's single-hop, Multi-Fragment Reasoning -> LOCOMO's multi-hop and temporal, Aggregation -> LOCOMO's open domain and AggQA).

**Evaluation metric inconsistencies**: Original publications of compared models (Section 3.2) for which LOCOMO evaluations exist consistently report F1 and BLEU scores, while this paper relies solely on LLM-as-a-judge evaluation. Only Chhikara et al. (2025) reported LLM-as-a-Judge scores. Notably, relative LLM-as-a-judge scores between models show near-zero correlation with scores reported by Chhikara et al. (e.g., for open-domain: Chhikara reported Zep=77, A-mem=54; this paper reports Zep=39, A-mem=50; for multi-hop: Chhikara reported Zep=41, LangMem=48; this paper reports Zep=14, LangMem=32). This suggests the metric is unreliable or should be complemented by additional metrics such as F1 or BLEU.

**Questionable importance of core mechanism**: Figure 5 demonstrates that changing $\theta_{meta}$ from 0.85 to 0.95 causes substantial behavioral changes (shift from assimilation to schema creation in Figure 5b), yet evaluation scores remain relatively stable (Figure 5a). This suggests the schema/bucket organization may not be critical for task performance.

**Missing computational efficiency analysis:** The authors never report computational performance metrics (e.g., inference time, memory consumption, number of LLM calls, retrieval latency) of PISA compared to baseline methods.

**Incomplete experimental analysis**: The $\theta_{meta}$ hyperparameter sweep is only performed on the LOCOMO dataset. Ablation studies are incomplete: PISA without Initialization is only evaluated on AggQA's medical domain, and PISA without Retrieval only on AggQA's finance domain.

**Minor**:
- Code is not currently accessible from the provided link.
- LOCOMO's adversarial questions are introduced in the main text but not used for evaluation.

**Questions:**

- Table 1 shows that providing full context to the model outperforms all memory systems on the LOCOMO benchmark. How do the authors reconcile this result with the paper's central premise that memory systems should address the limitations of full-context approaches (line 42: "excessive historical information [...] risks exceeding the model's context window; [...] amount of irrelevant information contained may interfere with the model's judgment and decision-making")?

- What accounts for the observation that both PISA and the full-context baseline achieve superior performance on AggQA's medium difficulty level compared to the easy difficulty level?

---

### Official Review · Reviewer_hiM2 · 2025-10-29

**Soundness:** 2
**Presentation:** 3
**Contribution:** 2
**Rating:** 4
**Confidence:** 2

**Summary:**

This paper introduces PISA, a unified memory system to improve an agent’s adaptability to diverse tasks and enhance task-oriented decision-making. Inspired by cognitive psychology, the authors propose a schema-based memory representation that enables agents to actively construct, refine, and retrieve relevant knowledge. The framework demonstrates improved performance in memory-dependent agent tasks, and the authors further design and evaluate the AggQA benchmark to assess task-oriented memory management. The results indicate that PISA offers advantages in organizing and utilizing memory for effective task execution.

**Strengths:**

- **S1**. The authors propose a schema engine that supports a constructive and adaptive hybrid memory system, allowing more structured and flexible memory management.
- **S2**. The design of the AggQA task effectively demonstrates the task-oriented properties and practical advantages of the proposed PISA framework.
- **S3**. Including ablation studies and evolutionary threshold analyses provides valuable insights into the contributions and behavior of different model components.

**Weaknesses:**

- **W1**. The criteria and methodology for defining the Meta, Element, and Element-Value categories remain ambiguous.
- **W2**. The ablation study results show that initialization is critical, and the system depends heavily on it. However, there is insufficient explanation regarding how initialization is performed using prior knowledge and how the schema configuration may vary across different domains.
- **W3**. In the PISA framework, temporal relationships and identity tracking for similar entities (e.g., Dog1 vs. Dog2) are not adequately addressed, limiting clarity on how memory maintains continuity over time.
- **W4**. The adaptive processing modules introduce schema-level similarity scores for schema matching and element-level compatibility scores for element matching, yet the operational details of these computations are not fully described. More elaboration is needed as these components appear essential for memory management within the PISA framework.

**Questions:**

- **Q1**. How is the boundary between Meta and Element determined, and what prevents overly broad or excessively fragmented schema definitions?
- **Q2**. Following Q1, are the keywords used for Meta, Element, and Value guaranteed to be unique? If so, how does the system handle cases where a keyword that appears as a Meta category should also appear as an Element or Value in a different context?
- **Q3**. How does the framework address temporal dependencies—such as ordering and causality—during memory retrieval and decision-making?
- **Q4**. What mechanisms ensure unique identity tracking when multiple instances share the same Meta category (e.g., two different entities both categorized under “Dog”)?

---

### Official Review · Reviewer_vGET · 2025-10-31

**Soundness:** 2
**Presentation:** 4
**Contribution:** 2
**Rating:** 4
**Confidence:** 4

**Summary:**

The authors of this paper propose a unified memory system for AI agents. The system is based on the concept of *schemas*, from the Piaget's schema theory. A schema encodes a structured knowledge, containing a meta (drink, music, ...), an element (coffee, jazz...), the value linked to the element (e.g. coffee details and preferences) and an experience identification.

The system itself proceeds in three steps: in a first step, the schemas are initialized based on existing data; then the schemas are updated over time continuously, with the possibility of assimilation (new values of an element), evolution (new element of a meta) or creation (new meta). At that stage, a conflict analysis is also performed. Finally, the system contains a retrieval module, that can query the memory pool with different tools (RAG, SQL, calculator).

The system is evaluated on LOCOMO and AggQA. AggQA is a new benchmark proposed by the authors, containing 134 questions from financial and medical domains (example of question: "What was the average closing price of Pinterest stock from April 1 to April 30, 2024?"). The evaluation is performed with gpt-5-mini against existing competitors (Mem0, LangMem, etc.) and direct answer by the model using full context. On LOCOMO, PISA shows better results compared to existing competitors, but is not better than full context. On AggQA, PISA outperforms both competitors and direct full context prompting.

**Strengths:**

- the concept of schemas is well explained and gives a structured way to represent the memory history, with some flexibility in creating and adapting memories
- consistent improvement against other memory retrieval methods on locomo (excluding the adversarial category)
- I checked the availability of both source code and data

**Weaknesses:**

- schema management seems simplistic for real cases (see questions below)
- the introduction of schemas does not seem to be the main component in the good performance
- the experiments do not clearly show the improvement compared to full context

**Questions:**

## Justification that the use of schemas is flexible and provides a significant improvement

The use of schemas is the central component of the system and of the paper.

- Q. Can you justify that the improved performance with the PISA system is because of the use of schemas instead of the retrieval module? In particular, how similar are the retrieval tools of the competitor methods?

The schema modelization seems simplistic. For formulating this issue objectively, can you explain how the mechanisms interact in those cases:

- Q. From example in Fig. 6. What about "The cat of my neighbor has 3 legs"? This may be true for this particular cat. How it should be handled ideally (the "cat" schema is updated? a new specific schema is created?)

- Q. From example in Fig. 1. What about "I like coffee with milk only when the music is jazz". How this is integrated?

- Q. Is it possible to update multiple schemas at once?


## Comparison with full context retrieval

Authors identify two challenges using AI agents: "risk of exceeding context window" and "irrelevant information may interfere with the model judgment". While I agree with the authors, the experiments against the full context size are not convincing on this aspect: comparing Tab. 2 and Tab. 3 on the AggQA financial dataset, without the retrieval module (fair comparison with full context), the results show that PISA (without the retrieval module) is not doing better than full context.

- Q. Can you confirm that in the full context experiment, no tool can be used

- Q. PISA is using both memory management and retrieval tools. Is it possible to compare in a fair way whether the proposed memory management is useful w.r.t. full context?

## Other questions

- Q. I don't understand clearly the initialization stage. Can you explain this phase in your experiment? What is the provided data at that stage?

- Q. why removing the adversarial category of locomo in Table 1?

- Q. what is the list of the memory pool buckets? In Fig. 1: user trait, user event, relationship; then in Fig. 3: user events, agent events. Is it fixed at the initialization stage?

- Q. How important is the conflict detection module in the experiment?

---

### Official Review · Reviewer_NCxp · 2025-10-31

**Soundness:** 2
**Presentation:** 2
**Contribution:** 1
**Rating:** 2
**Confidence:** 4

**Summary:**

the paper proposes PISA, a task oriented memory system for LLM agents, which is an active research subject

the paper casts a shade of theoretically principled design (by pompously recalling Piaget & Cook -- already erroneously cited as Piaget, Cook *et al.*, as if there was a third unkown coauthor) onto a
quite practical  implementation of a memory structured around schemas, that can be assimilated/accomodated (adding values into existing ones) or created ex novo (for new concepts).

the paper shows relatively simple experiment on one public (LOCOMO) and one purposely  built benchmark.  in short, the paper is lightweight in contribution and evaluation, and  overall contribution seems quite thin



Jean Piaget, Margaret Cook, *et al.* The origins of intelligence in children, volume 8. International universities press New York, 1952

**Strengths:**

- memory in LLM is an active topic
- the paper is clearly written

**Weaknesses:**

(detail of weaknesses expanded below in question section)

- the paper has lightweight contribution
- evaluation is lightweight and quite superficial
- evaluation benchmark appears lower than what publicly reported

**Questions:**

## Details about weaknesses

- the paper has lightweight contribution

In PISA Initialization/adaptation and retrieval module manage  memories as schemas. while the paper compares PISA to mem0, AMem and LangMem, the comparison remains superficial -- i,e., it is difficult to appreciate the "Piaget-inspired adaptive memory evolution mechanism,'' as well as what is the key that from  "[...] Piaget’s cognitive theory, that constructs task-oriented memory structure''.

the explanation is conducted with pedagogic examples, that remain excessively simplistic and are not remotely connected to the proposed benchmarks.


- evaluation is lightweight and quite superficial

the paper shows relatively simple experiment on one public (LOCOMO) and one purposely  built benchmark.


the evaluation is conducted with either classic aggregated values (on which there may be a problem, see next)  or with again pedagogic examples (in the appendix)

yet, even digging tyhe appendix the examples remain excessively simplistic, and are not digging to any extent the usefulness of the proposed schemas -- and its adaptive evolutionary status

for instance, Reecord Reliability Scoring depends on numerous weighting parameters, whose tuning in real-life settings may be involved and in the paper is barely mentioned and not even discussed (so for sure no ablation ios carried over it) -- but this is just one example



- evaluation benchmark appears lower than what publicly reported

The results of compared baslines appear lower than the expectations as already publicly reported  (eg see https://www.memobase.io/blog/ai-memory-benchmark ), and the published benchmark often exceeds PISA on LOCOMO.


        Method 		Single      Multi	OpenDomain	Temporal	         Overall
        mem0              67.13	51.15	72.93		55.51		66.88
       LangMem		62.23	47.92	71.12		23.43		58.10
       Zep			74.11	66.04	67.71		79.79		75.14
       OpenAI 		63.79	42.92	62.29		21.71		52.90
       Memobase  	    70.92	   46.88       77.17 		85.05 		75.78



## Minor/Languange/Style
typos:
"Reproducability statement" -> reproducibility p10

### overloaded styles
the paper mixes syntactical use of “” with overloaded semantic implications: you should only use it for examples but not for implementation

 - implementation: “JSONB”  (2.1_ schema engine)
 - examples:  topic “Drink” and specific Element “Pure Milk” (same section)


the paper again mixes syntactical use of “” with overloaded semantic implications, incresing confusion -- either you use “”for precise defs, or for approximation and intuitive metaphores

- precise definition:  [...] structured unit of knowledge within the system, also called a “Schema.”  [...]  “Assimilation" is the process of
- approximation/metaphore: This makes PISA’s schemas consistent with Piaget’s definition: they are executable “mini-programs”  [...]
- examples: different breed of dog into the “dog” category

### mixed styles

the paper mixes styling --  choose one but mixing the two is not recommended

	{\texttt RULER} vs RULER
	{\texttt LOCOMO} vs LOCOMO

---

### Note · Authors · 2025-11-12

I have read and agree with the venue's withdrawal policy on behalf of myself and my co-authors.